# An Analysis of Food Accessibility of Mountain Cities in China: A Case Study of Chongqing

Yufeng He [1,2,3,4], Haixia Pu [5,*], Nianhua Liu [4], Yongchuan Zhang [6,*] and Yehua Sheng [1,2,3]

1   School of Geography, Nanjing Normal University, Nanjing 210023, China; 181301016@njnu.edu.cn (Y.H.); shengyehua@njnu.edu.cn (Y.S.)
2   Key Laboratory of Virtual Geographic Environment (Nanjing Normal University), Ministry of Education, Nanjing 210023, China
3   Jiangsu Center for Collaborative Innovation in Geographical Information Resource Development and Application, Nanjing 210023, China
4   Department of Geoinformatics-Z_GIS, University of Salzburg, 5020 Salzburg, Austria; nianhua.liu@stud.sbg.ac.at
5   Department of Tourism and Land Resources, Chongqing Technology and Business University, Chongqing 400074, China
6   Chongqing Smart City Institute, Chongqing Jiaotong University, Chongqing 400067, China
*   Correspondence: superpu@ctbu.edu.cn (H.P.); zhangyc@cqjtu.edu.cn (Y.Z.)

**Abstract:** Mountain cities are characterized by undulating terrain, complex road networks, and diverse road facilities, which makes accessing food more difficult than in cities with a flat terrain. This study proposes an enhanced two-step method based on the Baidu map service for the construction of supermarket–market–retail food sales architecture and for calculating food accessibility. The accessibility indices of seven major food categories (grains and oils, fruits, vegetables, seafood, meat, milk, and eggs) were calculated considering the principle of the fairest walking routes in Chongqing. The correlations between food accessibility and house price and house age in Chongqing were explored through local Moran's analysis and geographically weighted regression. The correlations illustrated the fairness of the distribution of food accessibility in Chongqing among the poor and rich. The experiments showed generally well-developed food accessibility in the main urban areas of Chongqing. However, accessibility to fresh fruits and vegetables lagged in newly built urban areas.

**Keywords:** food geography; food accessibility; public health; E2SFCA; geographic regression





## 1. Introduction

The food environment of a region has a significant impact on the dietary habits and lifestyles of residents and is one of the key determinants of their health and well-being [1]. Residents of areas lacking healthy food are often prone to outbreaks of diseases such as obesity, diabetes, and malnutrition [2]. The availability and accessibility of healthy food has become an issue of global concern [3–6].

One definition of a "food desert" is an area where human residents have limited access to fresh and healthy food or where healthy food is unaffordable for poorer people [7]. Although there are different definitions, food deserts generally have three characteristics [3,8,9]: (1) availability of healthy food that is lower than consumption capacity; (2) food that is a long distance from the population; and (3) a close relationship between food accessibility and socioeconomic inequalities. "Health stores" are stores that provide fresh vegetables, fruits, and meat and mainly include hypermarkets and fruit and vegetable stores [10]. Based on their characteristics, vegetable farms, markets and seafood markets, and specialty food stores in China can be classified as health food stores.

Many previous studies have applied a geographic information system (GIS) within the analysis of the accessibility of health food stores. Consequently, many models have been established for measuring accessibility of health food that consider the complexity of

transportation systems, spatiotemporal variations in drivers, and the dynamics of urban life [11]. Food accessibility assessment has been the most common method for food desert research [3,8,10,12,13].

Most previous related studies have analyzed regional differences in healthy food accessibility using a simplistic descriptive analysis or have explored the interconnections between food deserts and socioeconomic regions at multiple levels using least squares regression and spatial autocorrelation methods [11]. In fact, there is interaction between socioeconomic activities at different levels. The socioeconomic level of a neighborhood is not only related to the characteristics of the community in which it is located but reflected in the influence of high-level macro conditions, such as at the street or county levels [14]. Therefore, many studies have attempted to avoid the analytical bias associated with these two relationships by using multilevel linear regression (hierarchical linear additive model) to explore the relationships between different levels of social indicators and food accessibility [14,15]. Of course, these studies were based on detailed economic data.

The development of the economy and the unique terrain of Chongqing are also important considerations for analyzing food accessibility. The early development of Chongqing involved the introduction of numerous industries. This development of Chongqing played an important role in the larger-scale development of China to set the foundation for a comprehensive national industrial base dominated by heavy industry, involving the machinery, chemical, metallurgical, food, and textile industries [16]. The duality of the overall development of Chongqing is reflected in: (1) the large proportion of traditional agriculture and its lagging development; (2) the large gap between income and consumption levels; (3) the large gap in labor productivity between industry and agriculture; and (4) numerous migrant workers in the city.

The development of industrial structure follows an evolutionary path from agriculture to light industry, to heavy industry, and finally to high-tech industry [17]. The economic base of the region remains weak and in an initial industrialization process, with the establishment of a large machine industry. On the one hand, traditional agriculture and handicraft industries exist in the region along with some heavy industry and advanced technology industries. On the other hand, processing industries remain limited, and light industries are still in an intermediate evolutionary stage. Therefore, the economic structure of Chongqing shows a weak correlation between the urban industry and the rural economy. This relationship is responsible for the movement of many migrant workers into the city [18].

The urban development and layout of Chongqing have benefited from the convenient access to two rivers. The expansion of Chongqing has been constrained by its topography. The formation of a more complex irregular road network represents a spatial leap in the development of Chongqing. Urban areas of Chongqing have been developed in areas with relatively low topography and are relatively fragmented.

It is difficult to estimate the accessibility of food in Chongqing due to the imbalance of the tertiary sector based on the industrial economy and the isolated vertical traffic pattern of the city.

The aim of the present study was to identify the spatial food distribution pattern of the mountain city of Chongqing at the community level. GIS was used to identify the walking time required to access food under a realistic road network using a Baidu map and considering the temporal factors limiting the accessibility metric. The augmented two-step method was established as the distance between community centers and different types of stores. This method was used to calculate food accessibility indicators reflecting the distribution characteristics of food accessibility/desert in Chongqing. The spatial correlations between socioeconomic indicators based on house price/age and food accessibility indicators reflected the balance of food distribution in Chongqing. These indicators were used to identify the status of and objective reasons for the different distributions of food accessibility. In addition, suggestions are made for the development and planning of the city. The main contribution of the present study is a proposed effective solution for food

accessibility analysis suitable for mountain cities. In addition, the present study describes the food distribution of Chongqing and provides some suggestions for development of the city.

This paper has the following structure: Section 2 provides a brief description of related work in the field and analyzes the status and limitations of current food access analysis; Section 3 describes the data used and study area; Section 4 describes the methods used in the present study; Section 5 summarizes the experiment results and analysis and suggests development plans for Chongqing; and Section 6 provides a conclusion and discussion of further research needed.

## 2. Related Work

### 2.1. Research on Food Deserts

Early studies on food deserts focused on regions with a food deficit [7], areas with a lack of supermarket competition [19], and regions undersupplied with affordable food products [20]. The United States Department of Agriculture (USDA) defines food deserts as areas in which residents cannot afford nutritious foods or where these foods are not available. Many past studies have proposed that food deserts develop because of a lack of retail food venues resulting from poverty or poor transportation links [3–5].

Since supermarkets are stocked with a wide variety of food types, many past studies on food deserts have used the distribution of supermarkets as an indicator of food accessibility [10,13,21]. The present study considered the above studies, since markets and stores also offer food for purchase.

The study of accessibility to healthy food involves the use of traffic models and the spatial distribution of stores [8,22–24], service capacity of supply stores [3,25], distribution of the community transportation systems, and personal preferences [15,26]. Therefore, the results of these studies considered combined impacts by those factors [9,27]. Many studies have measured food deserts by counting the number of supermarkets or stores in the field [28]. Other studies on food deserts used a range of services of stores and supermarkets to evaluate food accessibility [4,24,25,29]. A cumulative opportunity index has also been developed to measure access to healthy foods [28]. More explicitly, the "service area" is determined by establishing a buffer zone prespecified by distance or travel time, following which the total numbers of supermarkets and large grocery stores (SLGSs) within that service area are calculated. Recent studies on food deserts have focused on geographic accessibility, with methods for measuring distance- or time-based food accessibility gradually becoming dominant in the field.

### 2.2. Accessibility Indicators

The accessibility indicator represents the ease at which food can be accessed between a residential address and the food store [30,31], with some early studies basing this measure on the distance between them as a straight line or Euclidean distance. The use of straight-line distances alone is insufficient to assess food accessibility because of different road network conditions and travel modes. The road network-based travel time evaluation method optimizes the accuracy of food accessibility based on different travel patterns [30].

Given the rate of private car ownership in the city and the convenience of public transportation, travel by private car [12,32] and by public transportation [33] are the two most common modes. Other studies have identified actual travel impedances (e.g., one-way streets, speed limits, restricted turns, and traffic volumes) [34,35]. There are both spatial and temporal dimensions to restrictions to food access [36,37]; Widener and Shannon [31] argued that travel time should be considered when analyzing the accessibility of healthy foods. Travel time related to the road network does not adequately consider the time spent by those using public transport during transport connection delays. Farber, Morang [38] measured the accessibility of healthy foods by comparing the length of travel at different time levels. Widener analyzed the spatial and temporal constraints on the accessibility of healthy foods [30]. Ravensbergen compared the weekday and weekend access to healthy

foods by children [39]. Although these efforts have contributed to the understanding of healthy food accessibility, modeling large urban road networks is a monumental task, which complicates the process of assessing urban food accessibility.

### 2.3. Food Accessibility and Economic Properties

Many studies have explored the relationship between access to healthy food and individual or community characteristics of poverty from a social inequality perspective [40–42]. Access to healthy food is generally lower in communities with high rates of poverty, low income, and high unemployment and in predominantly minority populations [40,43]. However, most studies involving surveys have simply compared accessibility between communities with different levels of poverty or relied on descriptive analysis. These approaches ignore two key points: spatial autocorrelation and multilevel socioeconomic interactions [44,45]. The clustering of retail stores can be more profitable (Sage, McCracken, and Sage 2013), as transaction costs borne by retailers are reduced. As a result, competition may increase in these clusters because of multiple businesses offering similar products, which may act as a driver for fragmentation of retail to maintain sustainable benefits [46]. Many studies of this phenomenon have applied the classical ordinary least squares linear regression (OLS). However, the use of this method can lead to spatial nonsmoothness and autocorrelation challenges when the assumption of homoskedasticity of OLS is not satisfied [47]. Geographically weighted regression (GWR) is the most widely used form of regression in these studies because of its advantages in characterizing spatially nonsmooth relationships [48]. GWR generates location-specific estimates for the correlations between the dependent and explanatory variables [49,50]. Therefore, empirical analysis, which does not consider these two issues, can produce biased estimates. These biased estimates can lead to misleading urban planning proposals. GWRs at different spatial scales can be used to further examine the social equality of access to healthy food.

## 3. Data and Study Area

Chongqing is in the transition zone between the Qinghai–Tibet Plateau and the middle and lower reaches of the Yangtze River Plain. This city has mountainous terrain with large topographic undulations and an average elevation of 400 m. The peculiarities of the topography of Chongqing often require winding routes to be followed between two locations. This results in a mismatch between straight-line distance and time cost and difficulty in reflecting actual food accessibility. Chongqing contains 26 municipal districts, 8 counties, and 4 autonomous counties. With consideration of the distribution of urban areas, the present study analyzed nine main urban areas in Chongqing, namely, the districts of Yuzhong, Jiangbei, Nanan, Jiulongpo, Dadukou, Shapingba, Yubei, Banan, and Beibei, shown as Figure 1.

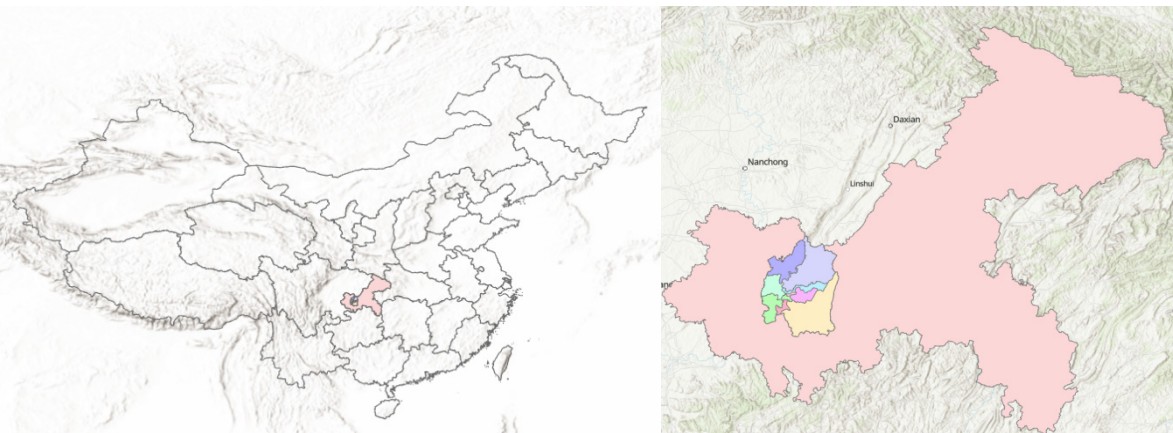

**Figure 1.** Map of Chongqing, China.

### 3.1. Food Type and Distribution

The food available in Chongqing can be divided into the major categories of grains and oils, vegetables, fruits, meat, eggs and egg products, milk, and seafood. An analysis of the types of commercial and store food sales in Chongqing indicated that the current main sources of food for sale were supermarkets, grocery stores, vegetable markets, farmers' markets, fruit stores, vegetable stores, seafood markets, seafood stores, milk stores, and butchers. Table 1 summarizes the goods offered by various stores.

**Table 1.** Food types and stores types in Chongqing.

| Food Type | Point of Sale | Weighting |
|---|---|---|
| grains and oils/egg production | Supermarkets, vegetable markets, farmers' markets, grain and oil stores, kiosks | 3, 3, 3, 3, 1/3 |
| Vegetable | Supermarkets, vegetable markets, farmers' markets, vegetable stores | 3, 2, 2, 2, 1 |
| Fruit | Supermarkets, vegetable markets, farmers' markets, fruit stores | 3, 2, 2, 2, 1 |
| Meat | Supermarkets, vegetable markets, farmers' markets, meat stores | 3, 2, 2, 2, 1 |
| Milk | Supermarkets, vegetable markets, farmers' markets, milk stores | 3, 2, 2, 2, 1 |
| Seafood | Supermarkets, vegetable markets, farmers' markets, fishery stores, crab stores, crayfish stores | 5, 3, 3, 3, 1, 1, 1 |

Since supermarkets can provide a wide variety of foods and serve the largest population, they were assigned a weight of 3. Vegetable markets and farmers' markets are traditional food vendors in China that sell all kinds of food. These markets are uniformly planned and managed by the government. The present study assigned a weight of 2 to these markets, because their capabilities are between those of supermarkets and detail stores in Chongqing. The market determines the locations of stores selling common vegetables, fruits, meat, egg products, and seafood, and the scope of service and capacity of these stores are relatively small. Oil and grains are widely available from small supermarkets or kiosks in Chongqing, which is of great convenience for the public. Therefore, the current study assigned a service capacity weight of 1 to these stores. The stores selling grains, oils, and egg products have similar distributions. The service ability weight for grain, oil and egg kiosks was 1/3. The weightings in Table 1 were formed after a field study of several relevant stores in the business district near Jiangbei campus of Chongqing Technology and Business University.

Various types of point-of-sale POI (point of interest) data were crawled in Baidu Map service, including data for 173 large supermarkets, 680 vegetable points of sale, 614 fresh aquatic product points of sale, 552 milk and dairy product points of sale, 623 meat points of sale, 5973 grain and oil points of sale, and 1249 fruit points of sale. These stores included supermarkets and markets (food market and farmers' markets). Figure 2 shows their spatial distribution in Chongqing.

### 3.2. Community Distribution and Housing Prices

Data for house prices were mainly obtained from Anjuke (anjuke.com) and 58.com (last accessed date: 2021-11-26). A total of 5213 community points and their average house prices were collected for the main urban area. The house prices in the main urban area of Chongqing ranged between 2897 and 37,290 RMB, with an average house price of 12,046 RMB. The main distribution of house prices ranged between 5000 and 20,000 RMB. Figure 3 shows the numbers of households in different communities. The results of local Moran's analysis are shown in Figure 4. Fields of "HousePrice"," BuildingYear" refer to Price and Building Year of communities separately. An analysis of the construction time of the communities showed a trend of continuous expansion to the city boundary. Figure 5 shows the distribution of community households, whereas Figure 6 shows price statistics information.

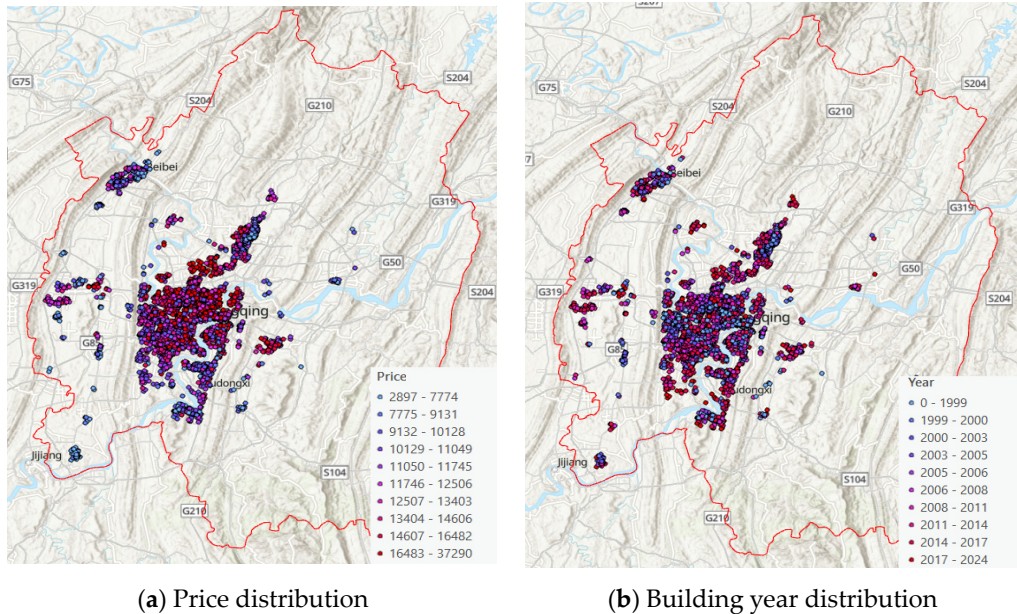

**Figure 2.** Spatial distributions of stores in Chongqing selling different categories of food products.

(**a**) Grain stores  (**b**) Vegetable stores  (**c**) Fruit stores

(**d**) Meat stores  (**e**) Seafood stores  (**f**) Milk stores

(**a**) Price distribution  (**b**) Building year distribution

**Figure 3.** The price and building year of communities in Chongqing.

(**a**) Price outlets        (**b**) Building year outlets

**Figure 4.** Local Moran's result for house price and building year in Chongqing.

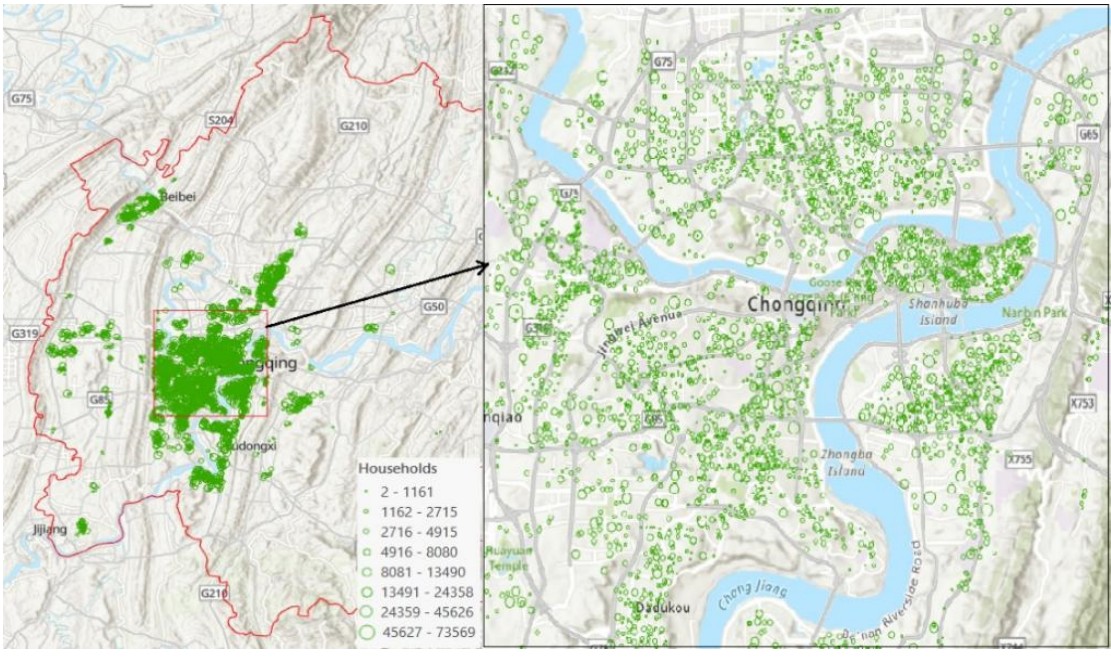

**Figure 5.** The households of communities in Chongqing.

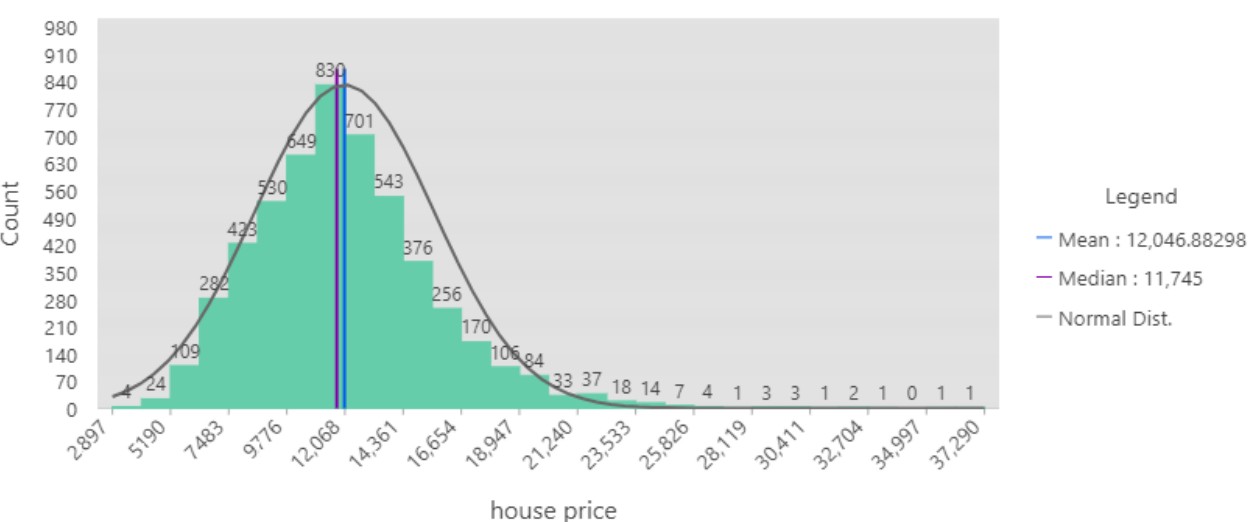

**Figure 6.** Distribution of house prices in Chongqing.

## 4. Methodology

### 4.1. Calculation of the Food Accessibility Indictor

#### 4.1.1. Travel Time Calculation

The rate of private car ownership in Chongqing is low. The undulating terrain also makes travel by bicycle uncommon. Access to public transport varies greatly between regions, complicating the accurate reflection of the food accessibility index. Walking is a common mode of travel and is more suitable for analyzing the accessibility of food necessities in Chongqing. Within the use of walking for analyzing food accessibility in Chongqing, categories of 15, 20 and 30 min walks can be applied. The present study applied a walk duration of 20 min as input to the model.

Traditional methods for calculating travel cost are usually based on complex road networks. Clearly, the representation of a large road network in a model is a huge barrier to calculating the time cost distance. With the calculation results considering a large road network, it can be difficult to consider uncertainties such as the real-time road network, road congestion, and road and bridge maintenance.

Baidu mapping services provide a real-time point-to-point time travel cost and path distance calculation service. Users of the service can rapidly calculate travel time and distance by setting a community center as the starting point and a shopping point as the target point through a Rest service request. This allows users to achieve real-time travel mode, travel time, and distance between any two points without the need to construct an urban road network model.

The present study used the Baidu mapping services to identify food stores within a 20 min walk (usually less than 1.350 m) of each community and calculated the time cost between the communities and the POI points of seven food types.

#### 4.1.2. E2SFCA for Calculating Food Accessibility

The two-step mobile search method (2SFCA model) considers both supply and demand factors, allowing for comprehensive and easy calculation of food store accessibility. E2SFCA-based food accessibility assessment is influenced by a combination of three factors: food stores, community population, and road network. The E2SFCA model avoids the potential dichotomy problem of 2SFCA by constructing a Gaussian decay model G. Figure 7 shows the conceptual description of the two-step mobile search method.

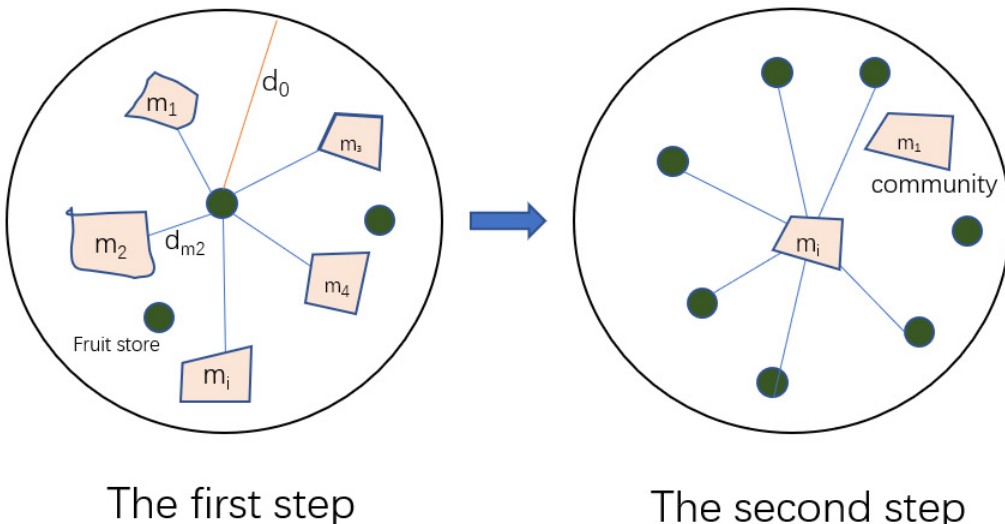

**Figure 7.** Conceptual description of the two-step mobile search method (E2SFCA).

Step 1: Given a food store POI location of type *k* as *j*, the model searches for any community *i* within the time cost distance threshold $d_0$ to form a region $F_j$ centered on *j*. The time cost value $d_{ij}$ between community $m_i$ and store $S_j$ is calculated by the Gaussian function G for weighting; the contribution of $C_j$ is accumulated for all communities *i* in the region $F_j$; and finally, the ratio $R_j$ between the population served ($Sev_j$) in store *j* and $C_j$ is calculated.

$$R_j = \frac{Sev_j}{\sum_{m \in \{d_{ij} \leq d_0\}} G(d_{ij}, d_0) * D_i} \tag{1}$$

Equation (1) assumes 2.45 people per household in Chongqing ($D_i = 2.45 \times Households_i$). The definition of $Sev_j$ is given in Table 1.

$$G(d_i, d_0) = f(x) = \begin{cases} e^{-\left(\frac{1}{2}\right) \times \left(\frac{d_i}{d_0}\right)^2}, & d_i \leq d_0 \\ 0, & d_i > d_0 \end{cases} \tag{2}$$

Step 2: For any store *j* within $d_0$ of distance from community *i*, the value of R weighted by the Gaussian function G is accumulated to obtain the accessibility value of *k* type of food at community *i*.

$$A_i = \sum_{m \in \{d_{ij} \leq d_0\}} G(d_{ij}, d_0) R_j \tag{3}$$

$A_i$ is a value that lies between 0 and 1. An $A_i = 0$ indicates that the community *i* is not reachable within a 20 min walk. The reachability of the community increases with increasing values of $A_i$. The algorithms were written to implement an enhanced two-step approach to calculating the accessibility index in combination with the Rest service (http://api.map.baidu.com/route=matrix/v2/walking?output=json). The accessibility indices were obtained under the same conditions by automatically obtaining the accessibility values of various types of food from 12:00 p.m. to 5:00 a.m. at night. Figure 8 shows the flow diagram of the algorithm. In this method, the first step uses the 1500 m range to filter out communities within 20 min for each store and then calculate the community's contribution (*r*) to the store. The second step uses the 1500 m range to filter the stores that can be reached within 20 min for each community and then calculates the comprehensive accessibility (*Ai*) by the r value for each community.

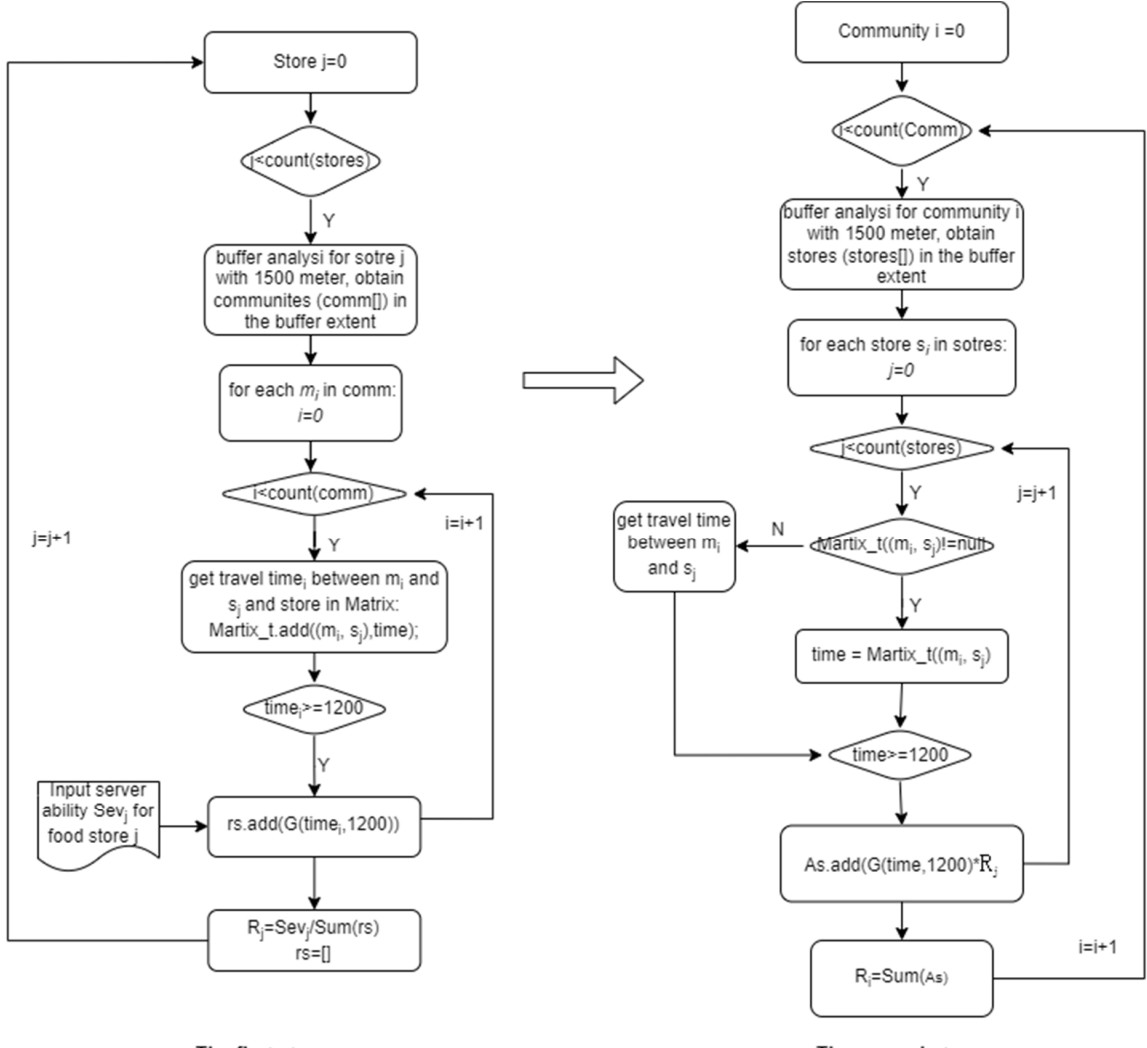

**Figure 8.** Enhanced two-step algorithm flow.

*4.2. GWR Regression Model*

House prices reflect to some extent the distribution of wealth in a city. An analysis of the spatial association between house prices and the accessibility indicator was based on the results of the analysis of the accessibility of various types of food in cities. This analysis can contribute to further strengthening the equity of food accessibility of cities. In addition, an exploration of the relationship between the construction time of communities and accessibility can reveal how the pattern of evolution in urban food accessibility is affected during urban development.

Brunsdonet [51] proposed the geographically weighted regression (GWR) method to study food accessibility. GWR is based on the idea of local smoothing and utilizes samples around the center for auxiliary regression to overcome the problems caused by nonsmooth spatial processes. Under GWR, the regression coefficients are generally considered to be spatially smooth to a certain degree and vary smoothly. The vicinity of the centroid can be regarded as approximately stationary, since the similarity of locations is inversely proportional to the distances between them, consistently with the first law of geography. The geographically weighted regression model is:

$$y_i = \beta_{i0} + \sum_{j=0}^{m} \beta_j(u_i, v_i)x_{ij} + \varepsilon_i \tag{4}$$

In Equation (4), $\beta_{i0}$ is the intercept coefficient at location $i$, $y_i$ is the dependent variable; $x_{ij}$ is the $j$th predictor variable; $\beta_j (u_i, v_i)$ is the $j$th regression coefficient; and $\varepsilon_i$ is the random error term [52].

$$\hat{\beta}(i) = \left[ X'W(i)X \right]^{-1} X'W(i)y \tag{5}$$

In Equation (5), $X$ is an explanatory variable by $k$ matrix, $W(i) = Diag[w1(i), \dots ,wn(i)]$ is a diagonal weight matrix in which weights are based on the distance between observation and location $i$, $\hat{\beta}(i)$ is the coefficients vector with $k$ by 1, and y is a $k$ by 1 observations vector of the dependent variable.

## 5. Result and Discussion

### 5.1. Food Accessibility Results

The present study calculated the accessibility values of different foods based on the enhanced two-step algorithm. Figure 9 shows the spatial distributions of accessibility of different types of food, with the point size in the figure representing the accessibility capability. Among the nine main urban areas in Chongqing, areas with high food accessibility were generally distributed in the center of each district. In Figure 9, the black dots represent the districts that were not accessible within a 20 min walk ($AI = 0$). The results showed no areas in the city that were inaccessible for grains, oils, eggs, vegetables, or fruits, whereas there were a few areas along the borders of each district or in newly developed areas that were inaccessible for seafood, meat, and milk. This result could mainly be attributed to many of the newly developed neighborhoods having incomplete facilities. In addition, there was a relative lag in access for some foods within a 20 min walk. Moreover, rural areas are often found at the periphery of urban areas; households in these rural areas raise some poultry, which was not considered in the present study.

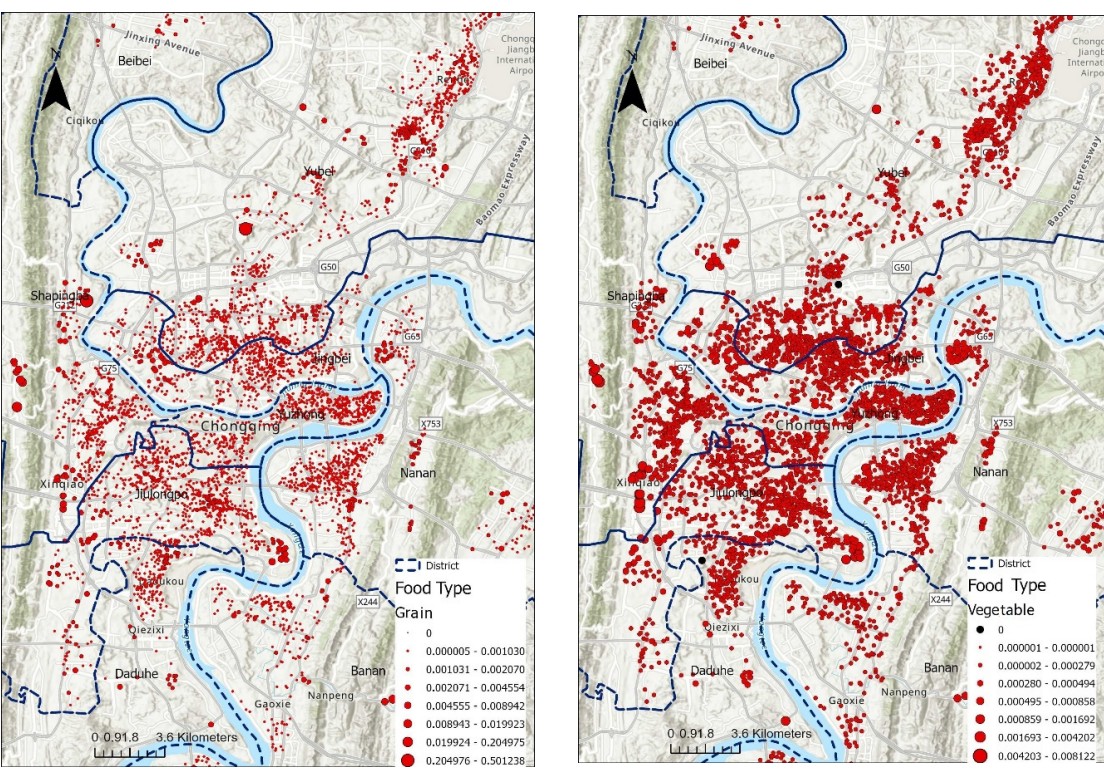

(**a**) Accessibility of grains, oils, and eggs      (**b**) Vegetable accessibility

**Figure 9.** *Cont.*

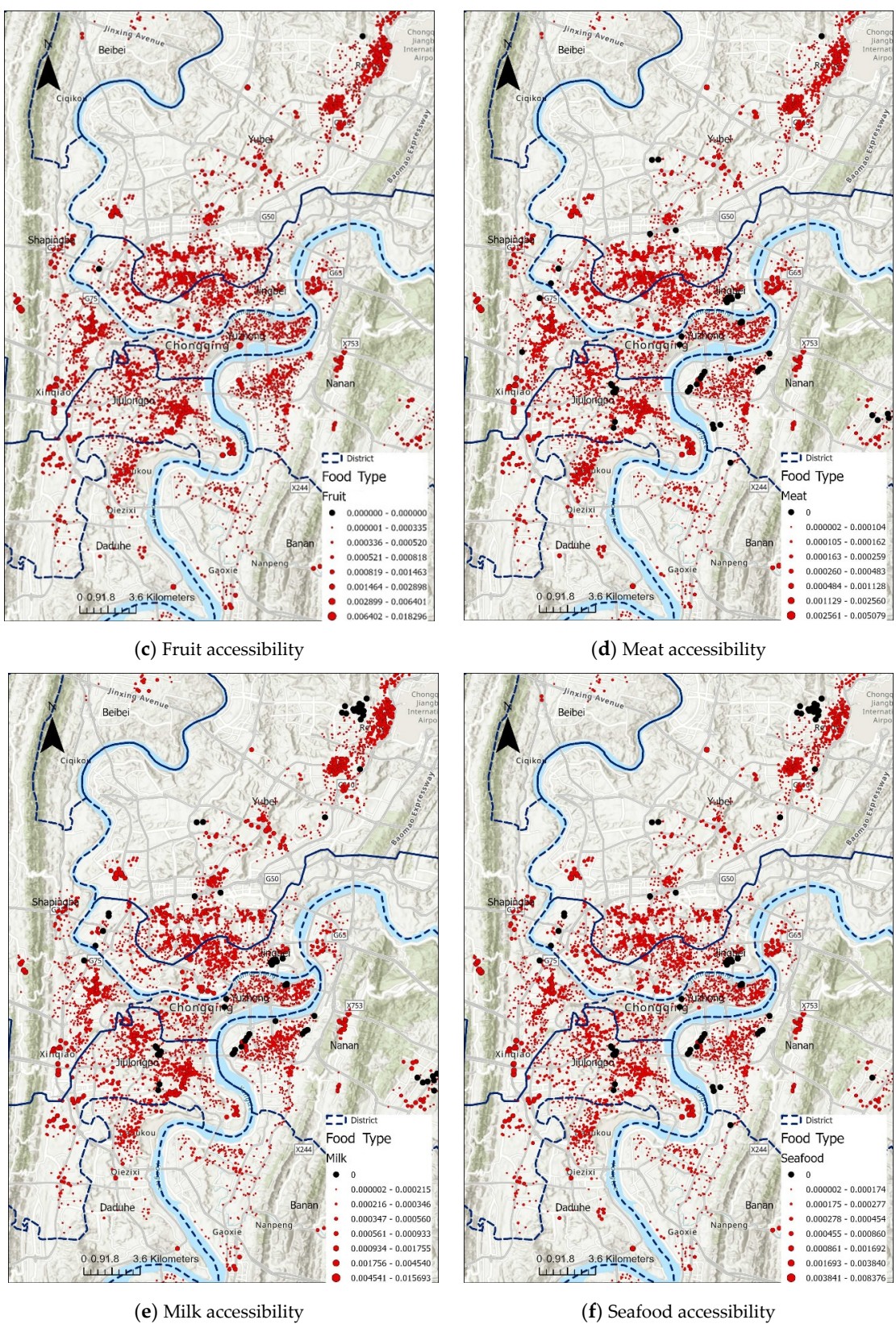

(**c**) Fruit accessibility

(**d**) Meat accessibility

(**e**) Milk accessibility

(**f**) Seafood accessibility

**Figure 9.** Spatial distribution of food accessibility in Chongqing.

The AI values of various food categories were categorized and plotted as box-and-whisker plots (Figure 10). When combining all types of accessibility indicators, the accessibility of grain and oil exceeded those of several other food categories, reflecting the high

accessibility of staple foods in Chongqing. The food category of fruits and vegetables had the second-highest accessibility, whereas the accessibility values of seafood, meat, and eggs were relatively small. These results are consistent with the larger situation in China of more affordable fruits and vegetables and lower demand for meat. Figure 10 was not used to filter the outlets.

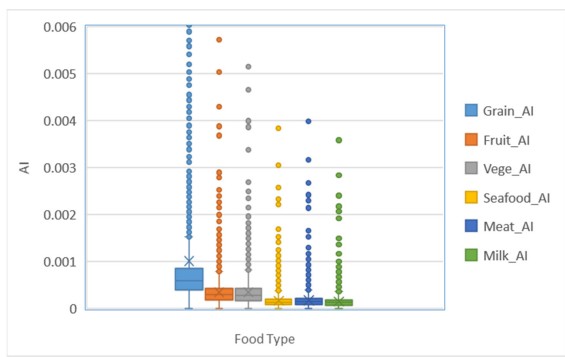

**Figure 10.** Accessibility indicator of different food types in Chongqing.

## 5.2. Correlation Analysis of Accessibility of Various Types of Food

The distribution of food and oil categories was generally even, with no obvious land aggregation evident (Figure 11), and there was good access to food and oil in the city. There were similar aggregations in accessibility for vegetables, fruits, meat, seafood, and milk. Food accessibility in each district gathered in roughly the same places, showing a strong correlation with the distribution of supermarkets and markets in the city. This result also reflected the high ability of the market to regulate food accessibility. Since the location of the market is controlled by the government, it also reflects the government's ability to control the accessibility of main foods, thereby facilitating the equitable management of the city.

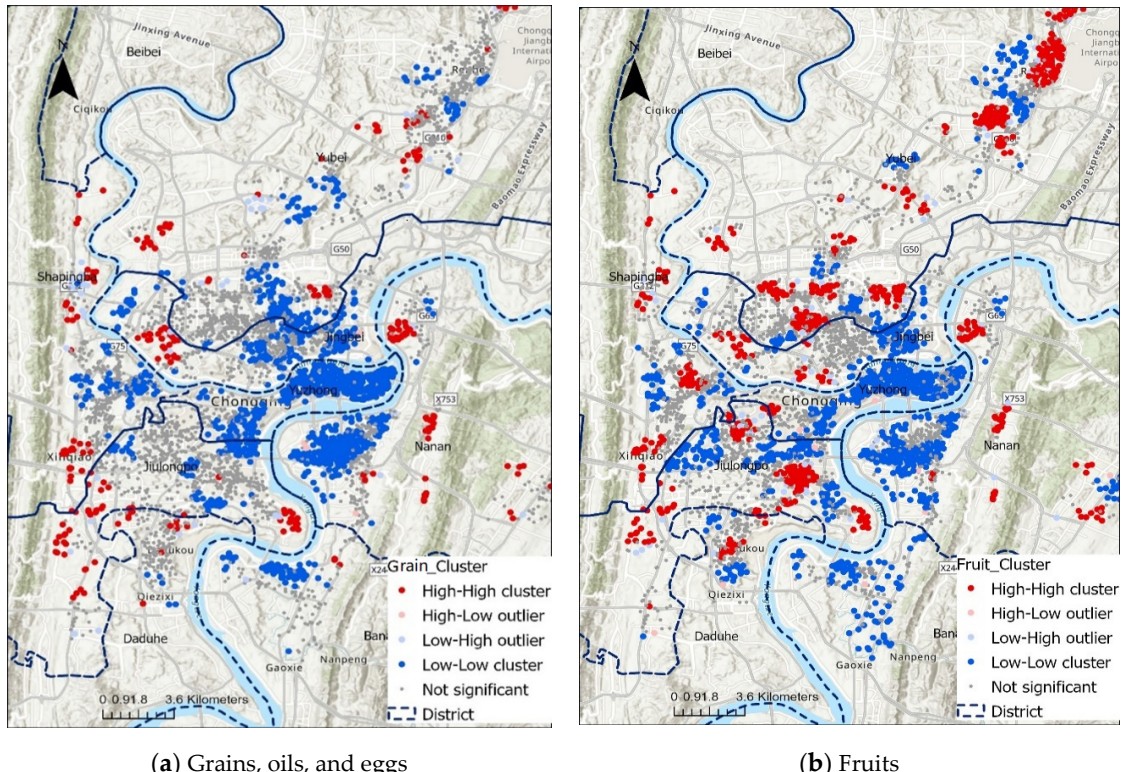

(**a**) Grains, oils, and eggs                    (**b**) Fruits

**Figure 11.** *Cont.*

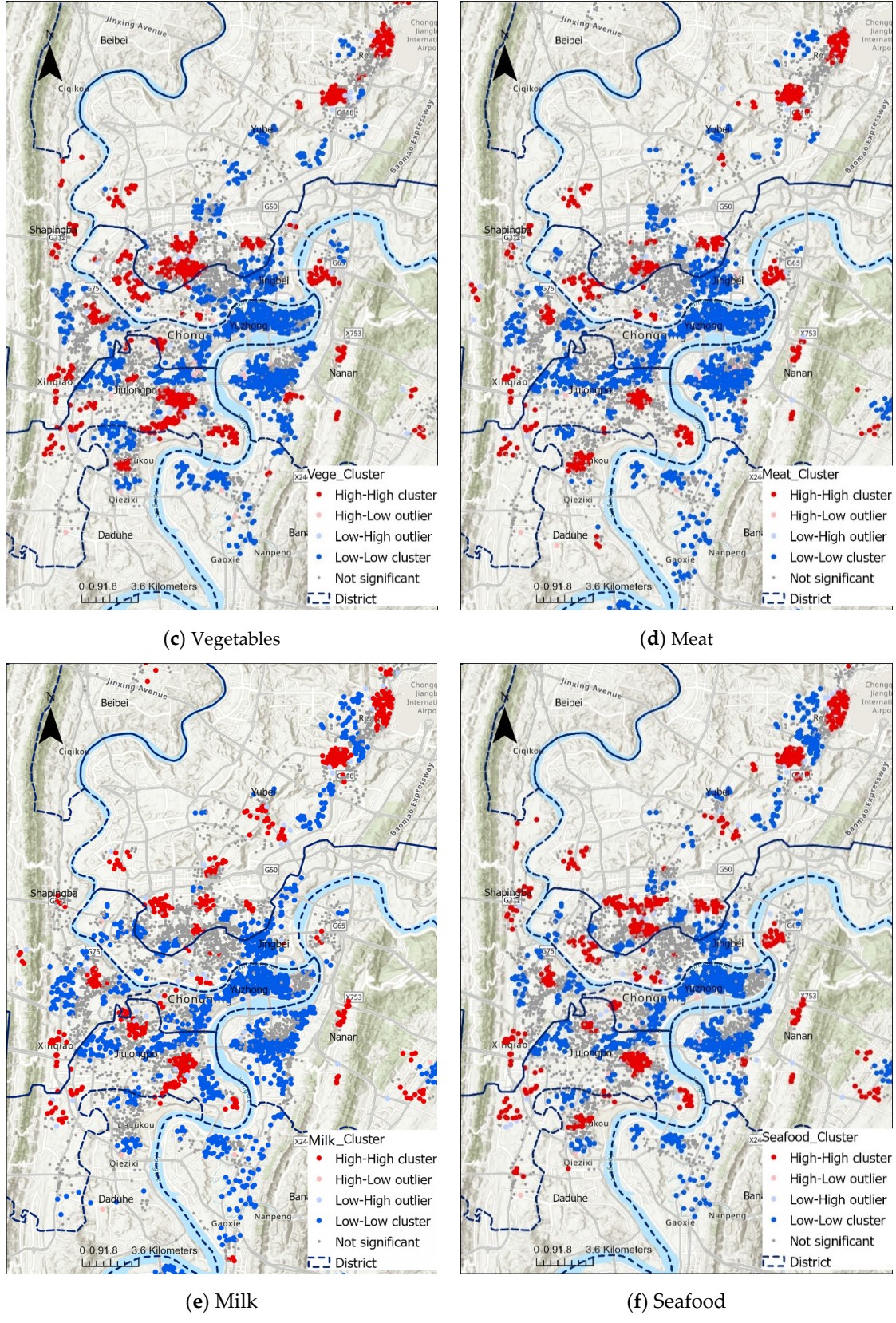

(**c**) Vegetables      (**d**) Meat

(**e**) Milk      (**f**) Seafood

**Figure 11.** Local Moran's analysis of different food types in Chongqing.

### 5.3. Results of Food Balance Analysis

5.3.1. Results of OLS Regression Analysis

The association between food accessibility and house price and house age was analyzed using OLS, with the results summarized in Table 2.

**Table 2.** The results of ordinary least squares (OLS) regression analysis.

| Food Type | Variable | Coefficient | *p*-Value | VIFc | Include/Omit | Adjust R2 | Koenker Test | AICc |
|---|---|---|---|---|---|---|---|---|
| Grian | Intercept | 0.000657 | 0.882179 | —— | | 0.001468 | 0.124751 | −33,027.596035 |
| | YEAR | −0.000002 | 0.002599 * | 1.002016 | include | | | |
| | LGPRICE | 0.000367 | 0.429808 | 1.002016 | omit | | | |
| Vegetable | Intercept | 0.001154 | 0.000000 * | —— | | 0.025729 | 0.000000 * | −71,557.891091 |
| | YEAR | 0 | 0.000000 * | 1.002016 | Include | | | |
| | LGPRICE | −0.000084 | 0.000000 * | 1.002016 | Include | | | |
| Fruit | Intercept | 0.001757 | 0.000000 * | —— | | 0.010771 | 0.108091 | −65,365.726053 |
| | YEAR | −0.000000 | 0.012975 * | 1.002016 | include | | | |
| | LGPRICE | −0.000139 | 0.000000 * | 1.002016 | include | | | |
| Seafood | Intercept | 0.000967 | 0.000000 * | —— | | 0.014320 | 0.004605 * | −72,451.277216 |
| | YEAR | −0.000000 | 0.000975 * | 1.002016 | Include | | | |
| | LGPRICE | −0.000078 | 0.000000 * | 1.002016 | Include | | | |
| meat | Intercept | 0.000664 | 0.000000 * | —— | | 0.017188 | 0.000855 * | −77,181.156384 |
| | YEAR | −0.000000 | 0.000514 * | 1.002016 | include | | | |
| | LGPRICE | −0.000054 | 0.000000 * | 1.002016 | include | | | |
| Milk | Intercept | 0.000847 | 0.000000 * | —— | | 0.003862 | 0.640117 | −68,671.957804 |
| | YEAR | −0.000000 | 0.587469 | 1.002016 | omit | | | |
| | LGPRICE | −0.000066 | 0.000006 * | 1.002016 | include | | | |

As shown in Table 2, the accessibility values of the six food categories were not significantly associated with house price or house age. The * of *p*-value represents significant correlation. House price was not significantly correlated to accessibility of grain and oil. However, there was a significant correlation between time of house construction and accessibility of grain and oil. This relationship requires further exploration.

There were significant correlations between the accessibility values of vegetables, fruits, seafood, and meat and both house price and house age. However, the fits of the OLS model to these relationships were not high. In addition, Koenker's test for heteroskedasticity indicated a regional distribution of both relationships. And the * means that it was required to detect spatial heterogeneity by GWR. House price was significantly correlated to milk accessibility, whereas house age was not significant.

5.3.2. Geographically Weighted Regression Analysis

Using food accessibility withing a 20 min walk as a spatial range, the present study used a geographically weighted regression of the valid variables retained by OLS using a broadband of 1200 m. Table 3 shows the GWR model fits (R2 values) for various food items. The results indicated that vegetables, fruits, seafood, meat, and milk show significant aggregation in different areas. Since walking 1200 m takes approximately 20 min, the choice of 1200 m as the bandwidth of GWR was beneficial to local regression.

**Table 3.** Results of geographically weighted regression analysis.

| Food Type | Variable | Bandwidth | AICc | R2 | Adj.R2 |
|---|---|---|---|---|---|
| Grain | Year | 1200 | 12,899.362 | 0.375 | 0.342 |
| Vegetable | Lgprice, Year | 1200 | 8779.114 | 0.990 | 0.990 |
| Fruit | Lgprice, Year | 1200 | −4477.295 | 0.978 | 0.977 |
| Seafood | Lgprice, Year | 1200 | −6846.700 | 0.986 | 0.985 |
| Meat | Lgprice, Year | 1200 | −11,478.409 | 0.994 | 0.994 |
| Milk | Lgprice | 1200 | 1665.625 | 0.926 | 0.923 |

GWR obtained an overall good fit that was far superior to that of OLS. The coeffects of all variables with a *p* < 10% were plotted, following which they were divided into two categories: greater than 0 and less than 0. For house price, a coeffect of house price (beta_igprice) value greater than 0 represents increasing food accessibility with increasing house price, whereas a beta_igprice value less than 0 represents the opposite. For house age, a coeffect (beta_Year) value greater than 0 represents decreasing food accessibility with increasing age of the neighborhood, whereas a beta_Year value less than 0 represents the opposite.

As shown in Figure 12a–e, the accessibility values of vegetables, fruits, seafood, meat, and milk increased with increasing house prices in some old urban areas. Furthermore, lower food accessibility corresponded with increased housing prices in some new development areas far from the center of the city (University Town, Airport New District and Northern New Town, etc.). On the one hand, newly developed areas contain a higher number of new houses. These new houses have been more expensive than older houses since 2018 because of the restrictions of Chinese real estate policy and a higher demand for new houses. On the other hand, vegetable street markets exist in suburban areas, and some residents of these areas grow their own vegetables, thereby further enhancing food accessibility. This is related to the more mature food supply chain of the old city; that of the new city is expected to gradually develop.

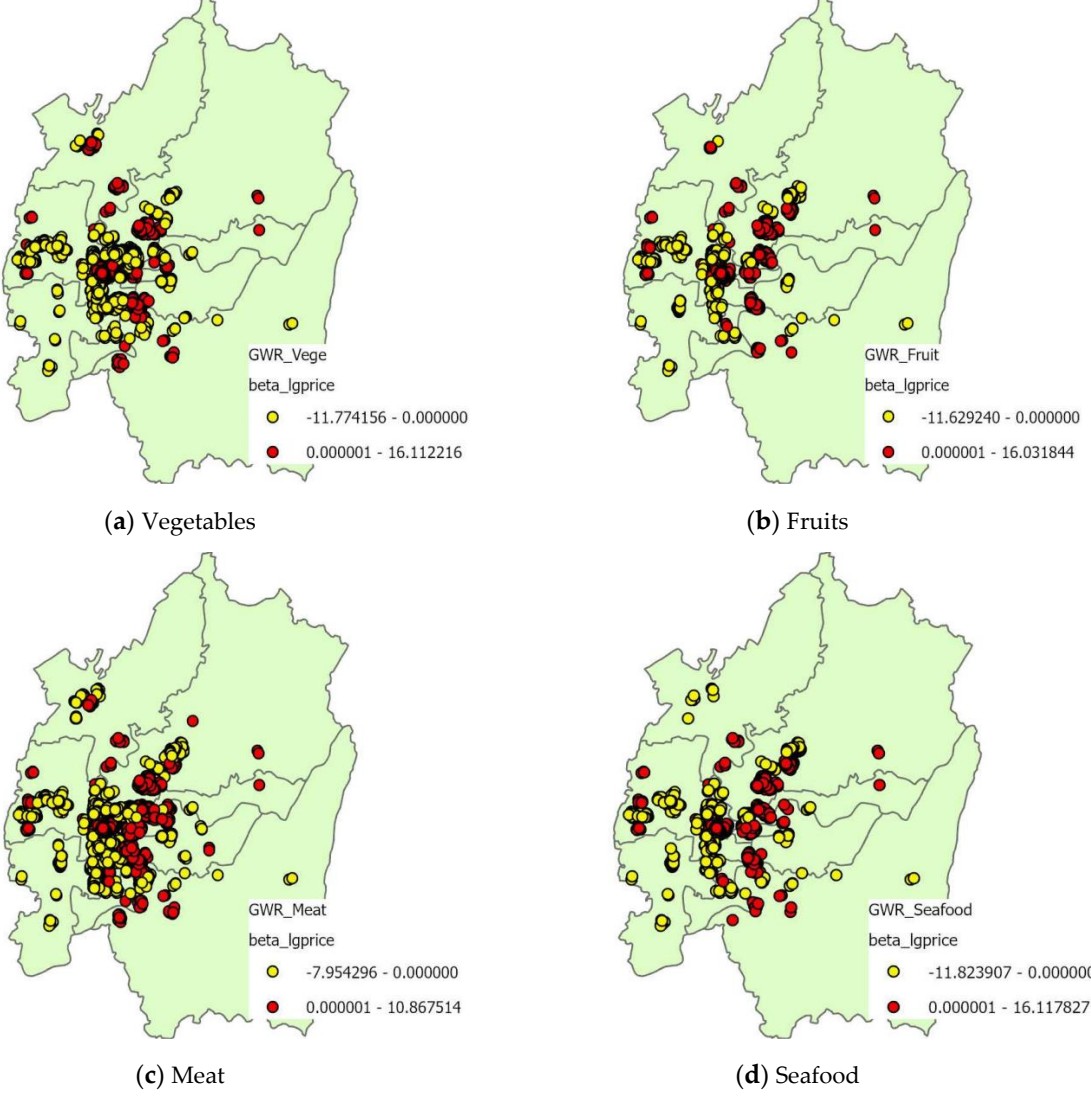

(**a**) Vegetables

(**b**) Fruits

(**c**) Meat

(**d**) Seafood

**Figure 12.** *Cont.*

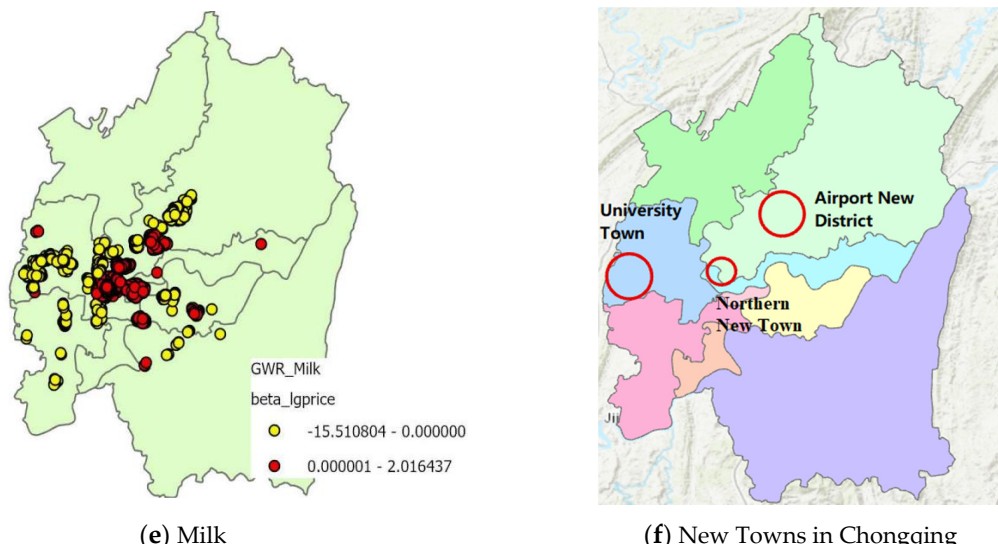

(**e**) Milk

(**f**) New Towns in Chongqing

**Figure 12.** House price coeffects with the accessibility values of vegetables, fruit, seafood, meat, and milk in Chongqing.

Although the areas of higher accessibility to grains, oils, and eggs were aggregated in regions with higher house ages, the highest accessibility in these areas corresponded with relatively newer houses (Figure 13a). This result can be attributed to the influence of the school district in the old city on house prices, and there is generally a strong correlation between the prices of new houses in the old city and accessibility of food and oil. Accessibility of food and oil is not a determinant of house prices. Therefore, food and oil are easily available in the old city, regardless of location.

As shown in Figure 13a–e, there was a clear pattern between the age of houses and accessibility of grains, oils/eggs, vegetables, fruits, seafood, and meat. The results showed that accessibility of food in communities far from the city center increased with decreasing age of houses. The opposite pattern occurred in the town center. The different trends in food accessibility between the two development areas of University City and the Airport City can be attributed to the development of the University City having initiated in 2007, whereas the Airport City was developed earlier in 2001. University City only initiated its first major shopping area in 2018, with a massive nonstudent gradually moving into the area since 2019. In contrast, the Airport City has experienced progressive development since an earlier period. $R^2$ coeffects for different types of food were shown in Figure 14.

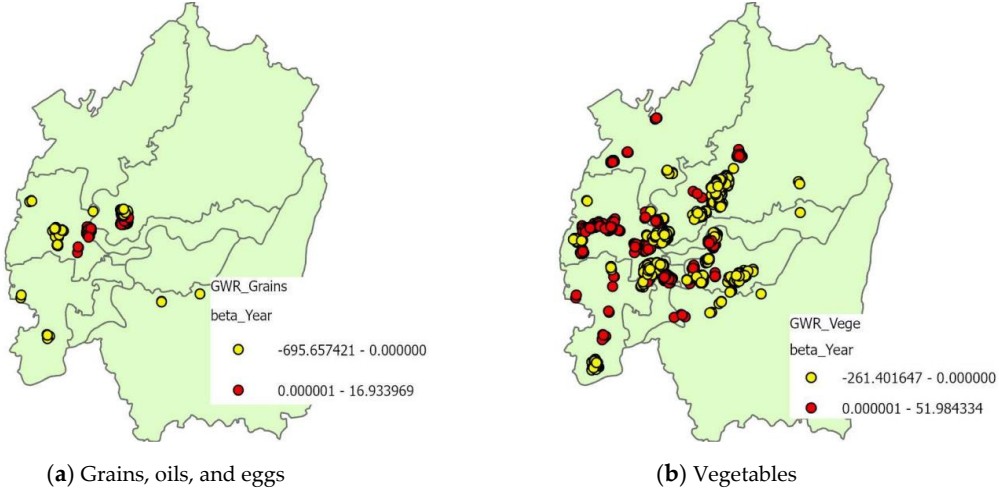

(**a**) Grains, oils, and eggs

(**b**) Vegetables

**Figure 13.** *Cont.*

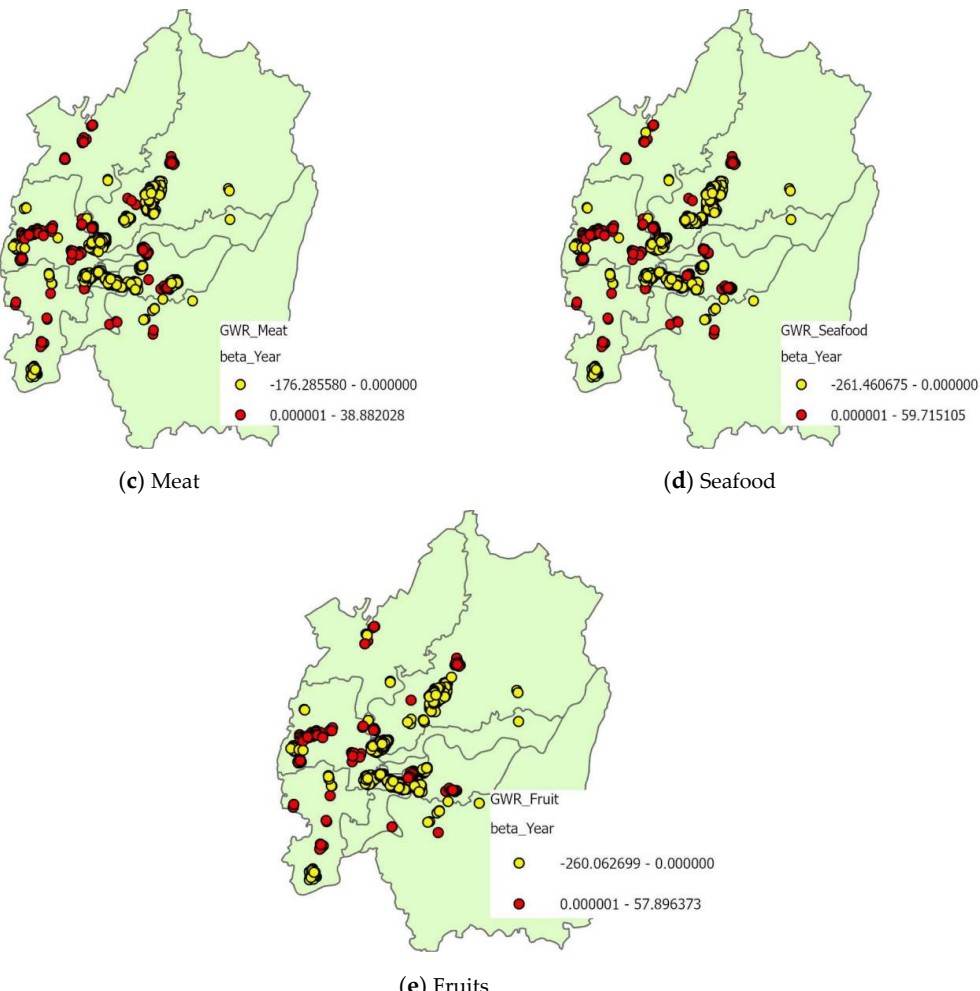

**Figure 13.** House building year coeffects with the accessibility values of grains, oils, and eggs; vegetables; fruits; seafood; and meat in Chongqing.

The concept of food accessibility originated in Western countries. Significant food desert areas are rarely found in China because of the relatively homogeneous urban population. The present study assessed food accessibility in the main urban area of Chongqing to reveal the accessibility of different types of food in the city. The results showed that the accessibility of oil in Chongqing exceeds that of vegetables and fruits, which in turn are higher than that of meat, milk, and raw food. The results of the present study are consistent with the dietary habits of Chinese families, in which there is a balance between the consumption of fruits/vegetables and meat. In general, besides a few fresh food and meat items, all food items in urban areas of Chongqing were accessible within a 20 min walk.

The present study examined food stores in the city of different sizes and considered the multitier service capacities of supermarkets, markets, and retail stores to reflect the accessibility of food more comprehensively in Chongqing. The zones of aggregated human habitation in Chongqing corresponded to higher accessibility to healthy food. From an economic perspective, the Jiangbei area showed significant clustering of areas with accessibility to healthy food. However, there were no significant differences in accessibility among the different categories of food. The results of GWR showed that the accessibility of seafood and meat increased with increasing house age in less affluent areas. Overall, there was no significant spatial change in the pattern of food accessibility. These results indicate that there is overall adequate and equitable accessibility to food in Chongqing. The current paper recommends that supermarkets or vegetable markets be promptly established for remote communities that may develop in the future.

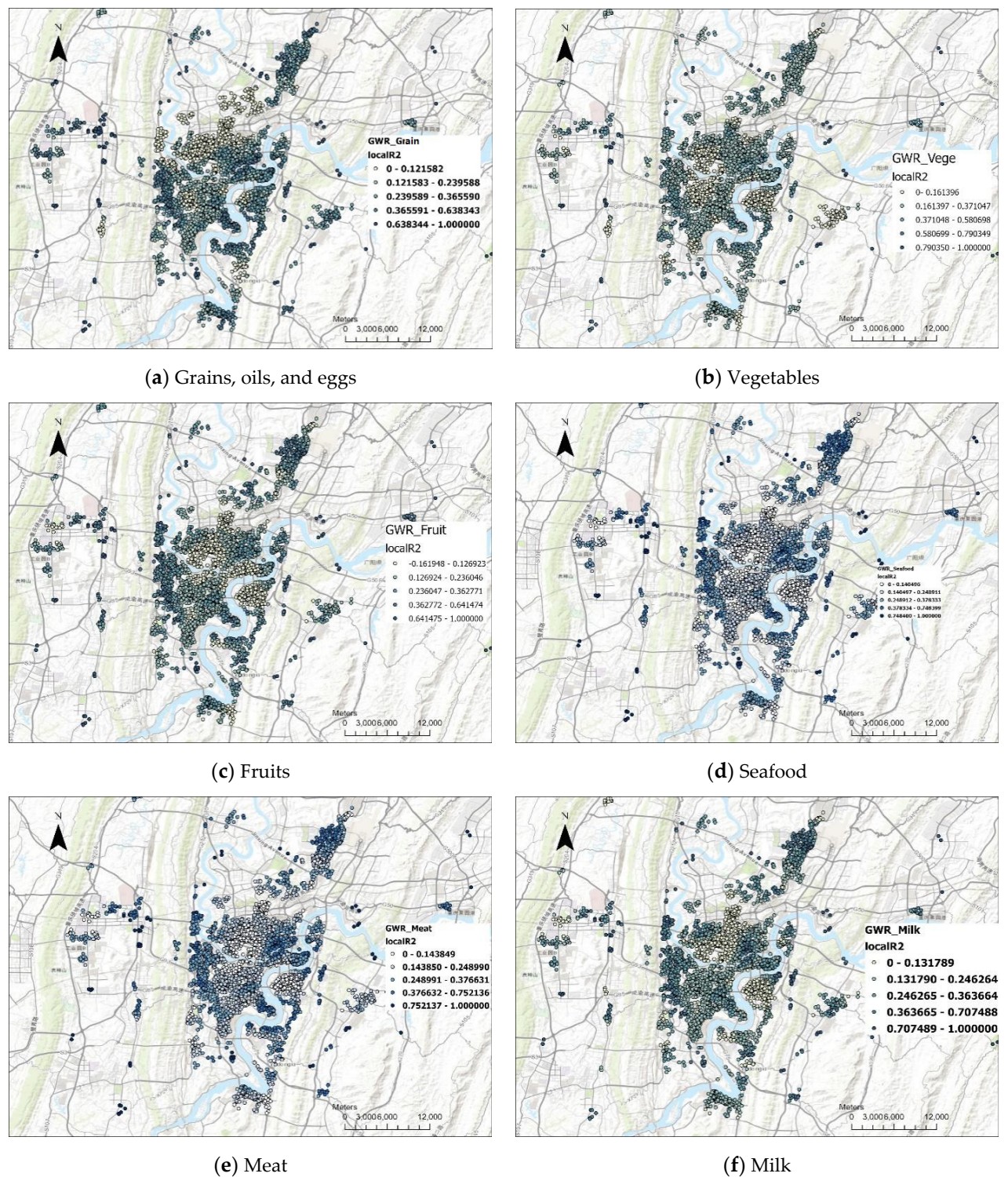

(**a**) Grains, oils, and eggs

(**b**) Vegetables

(**c**) Fruits

(**d**) Seafood

(**e**) Meat

(**f**) Milk

**Figure 14.** $R^2$ coeffects of grains, oils, and eggs; vegetables; fruits; seafood; meat; and milk in Chongqing.

## 6. Conclusions and Further Work

The present study constructed a food accessibility model based on the supermarket–market–retail structure and configured the service capacity of the model based on POIs and point-of-sale classification of food for sale in Chongqing. The enhanced two-step method was then combined with the Rest service based on Baidu's batch counting path to calculate the community food accessibility metrics algorithm. The relationships among

neighborhood house price, house age, and food accessibility were explored using local Moran's statistics and the geographically weighted regression method to reveal the social equity of food accessibility in Chongqing. The results showed generally good accessibility of the different types of food in Chongqing, besides low accessibility of meat and seafood in a few newly built neighborhoods far from the urban center. Meanwhile, an examination of the relationship between house price data and food accessibility indicated no significant inequity in food accessibility in Chongqing. There was a positive correlation between the age of a house and accessibility to seafood, meat, vegetables and fruits in the new city extension area. This result indicates the need for government agencies to update living facilities within newly built urban areas to ensure food accessibility for residents.

The assessment method in this paper is applicable to food accessibility analysis in all types of cities. Weightings for different stores can be set based mainly on the store turnover and consumer flow. Another method that refers to consumer flow can also be used to obtain the weights for different types of stores in the further work. We will also consider further using expert systems and hierarchical analysis methods to determine more accurate parameters in our future work. A 20 min walking time was used as a condition for food desert detection to take into account the inconvenience of transportation in Chongqing as a mountain city. For general cities, the assessment of the 15 min living circle range is of significance for research. The present study limited exploration to the regional balance of food accessibility and economic development in Chongqing from the perspective of house prices. Future studies could extend the current study to examine further correlations, such as community age distribution, employment status, literacy level, and gross domestic product (GDP).

**Author Contributions:** Conceptualization, Y.H. and Y.S.; data curation, Y.H. and N.L.; funding acquisition, Y.S.; investigation, H.P. and N.L.; methodology, Y.H. and H.P.; resources, Y.Z.; supervision, Y.S.; validation, Y.Z.; writing—original draft, Y.H.; writing—review and editing, Y.H. and H.P. All authors have read and agreed to the published version of the manuscript.

**Funding:** This research was funded by the National Natural Science Foundation of China: 41631175, 42077003; the Fund of Chongqing Social Science Planning Project: 2020QNGL39.

**Institutional Review Board Statement:** Not applicable.

**Informed Consent Statement:** Not applicable.

**Data Availability Statement:** The computational environment used during this research included an HP Zbook 17 laptop with a Windows 10 operating system, 16 GB RAM, and a 500 GB hard disk. The data mentioned (community points with price and age attributes, different kinds of store points) and the code used are available in a GitHub repository: https://github.com/Yufeng05He/FoodAccessAnalysis.

**Conflicts of Interest:** The authors declare no conflict of interest.

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
