# Peer review of "An Analysis of Food Accessibility of Mountain Cities in China: A Case Study of Chongqing"

_applsci, doi:10.3390/app12073236_

Round 1
Reviewer 1 Report
The main aim of the article is to identify the spatial food distribution pattern of the mountain city of Chongqing at the community level. The food accessibility is assessed by a two step method. Authors aim to answer if the suggested types of food are accessible equally throughout the city. The food accessibility is related to socioeconomic indicators based on house price and house age.
The article deals with an interesting topic. Paper contributes to the existing body of literature on food deserts. Authors explore the walking distance accessibility to the nearest supermarkets and markets. It would be even more interesting if authors could include the altitude and the effort to walk up and down, as the case study is a mountainous area.
The article includes the use of sound methods in spatial ekonometrics. The OLS is usually used to compare the GWR results, therefore I would recommend combining GWR and OLS tables together.
Minor comments:
- Figure 11 Local Moran’s analysis of different food types in Chongqing (e) Milk -> in the legend there is Seafood_Cluster instead of Milk_Cluster
- Line 415 “Table 3 shows the OLS model fits (R2 values) for various food items.” Table 3 shows GWR results
Reviewer 2 Report
The paper provides a framework to assess food availability in a geographical zone.
I found the paper clear and easy to follow.
The idea of considering both the closeness of shops to communities and that of communities to shops is interesting.
My only main comment is that the weights assigned to the different shop typologies feel a bit arbitrary. What would happen if you were to vary them slightly? Would the conclusions still hold, at least at a qualitative level? Maybe the authors could include a sensitivity analysis.
An alternative approach could be to obtain the weights by working with experts, using approaches such as the Analytic Hierarchy Process.
Figure 8 should be better described in the text of the manuscript.
In Figure 10 the distributions show several outliers. However, the outliers seem to be more or less uniformly spaced along the y axis. So this reviewer wanders whether these can really be treated as outliers and how would the result change when they were to considered as regular samples.
Maybe the authors could argue that, although seeming visually many, the outliers are comparatively much less than the regular samples used for the boxplot, if this is the case.
